# The Hong Kong version of Montreal Cognitive Assessment for the Visually Impaired (HKMoCA-VI): Proposed cut-off and cognitive functioning survey of visually impaired elderly in residential homes

**Calvin Chi Kong Yip[1], Winsy Wing Sze Wong[2]\*, Calvin Pak Wing Cheng[3], Armstrong Tat San Chiu[4]**

1 School of Medical and Health Sciences, Tung Wah College, Mongkok, Kowloon, Hong Kong, SAR, China, 2 Department of Chinese and Bilingual Studies, Faculty of Humanities, The Hong Kong Polytechnic University, Hunghom, Kowloon, Hong Kong, SAR, China, 3 Department of Psychiatry, School of Clinical Medicine, Lee Ka Shing Faculty of Medicine, The University of Hong Kong, Hong Kong, SAR, China, 4 The Hong Kong Society for the Blind, Hong Kong, SAR, China

\* winsyws.wong@polyu.edu.hk

**Data Availability Statement:** The data relevant to this paper are available from OSF at osf.io/62gwf.

## Abstract

### Background

Visual impairment has been strongly associated with the incidence of dementia. Appropriate cognitive screening for the elderly with visual impairment is crucial for early identification of dementia and its management. Due to challenges in processing visually presented stimuli among participants, the cut-off score of the Hong Kong version of the Montreal Cognitive Assessment for the Visually Impaired (HKMoCA-VI), also known as MoCA-BLIND or MoCA-22, was unknown. Besides, the cognitive status of elderly with visual impairment residing in care homes is rarely investigated. The current study aimed to 1) establish the cut-off score for HKMoCA-VI and 2) examine the general cognitive functioning of elderly with visual impairment living in residential homes in Hong Kong in terms of MoCA-VI percentile scores.

### Method

HKMoCA-VI and the Cantonese version of the Mini-Mental State Examination (CMMSE) were administered to 123 visually impaired elderly residents in care homes in Hong Kong. Percentile scores of HKMoCA-VI by age and education level were determined, and the concurrent validity, sensitivity, and specificity of HKMoCA-VI were assessed.

### Results

A cut-off score 12 was suggested for HKMoCA-VI, which yielded a sensitivity and specificity of 89.29% and 83.58%, respectively. Moreover, it strongly correlated with CMMSE, indicating satisfactory concurrent validity.

**Funding:** The author(s) received no specific funding for this work.

**Competing interests:** The authors have declared that no competing interests exist.

## Conclusions

HKMoCA-VI is suggested to be a viable cognitive screening tool for elderly individuals with visual impairment in residential homes. Further modifications to enhance the sensitivity and specificity of the measure are proposed.

## Background

Dementia, a degenerative neurological disorder affecting cognition, behavior, and ability to perform everyday activities, is a global public health concern [1]. Over 55 million people worldwide are affected by dementia, with 10 million new cases every year [2]. In Hong Kong, dementia affects approximately one-tenth of the elderly population [3]. The proportion of individuals aged 65 years or above in Hong Kong is projected to double from 16% in 2016 to 32% in 2046 [4], highlighting the urgency of preventive and management measures for dementia as the city transitions to a super-aged society.

Numerous risk factors for dementia, such as smoking, drinking, obesity, and depression, have been identified by the *Lancet* Commission Report [5]. Visual impairment has also been linked to increased cognitive decline in older adults [6–8]. In a cohort study examining 1,061 older women in the United States [9], participants with baseline visual impairment experienced a 2- to 5-fold increased risk of dementia over an average follow-up of 3.8 years, with risk escalating with severity of visual impairment. Similar findings were observed in studies conducted in Taiwan [10] and Hong Kong [11], further suggesting visual impairment as a potential risk factor for dementia. However, the underlying mechanism linking visual impairment to increased dementia risk remains unclear (e.g., see Discussion in Baltes and Lindenberger [12]). It has been hypothesized that visual impairment reduces participation in physical and cognitive activities [13], leading to neural degradation and cognitive decline in the long run. Moreover, a majority (57.1%) of the visually impaired suffer from multiple disabilities [14], potentially heightening their risk of dementia further.

Early diagnosis of dementia carries significant implications for persons with dementia, carers, and society as a whole. It not only aids in reducing associated risk factors but also helps delay disease progression [15]. Hence, valid and reliable cognitive screening tools have become vital. However, in the most widely used Montreal Cognitive Assessment (MoCA) [16], 8 out of 30 scores are attributed to items reliant on visual function, posing challenges in generalizing screening scores and percentiles to the population with visual impairment.

The MoCA 5-minute protocol [17], a validated tool for telephone administration, includes only the 1-minute verbal fluency task for assessing executive function, omitting assessment items related to attention and abstract concepts that rely on auditory function. Conversely, the Montreal Cognitive Assessment-Blind (MoCA-BLIND) or MoCA-22 [18, 19] is an adapted version of the original MoCA, specifically designed for the population with visual impairment, as it excludes items requiring visual abilities. In this study, we used the Cantonese-translated MoCA and eliminated items necessitating visual abilities in line with the MoCA-BLIND or MoCA-22. To distinguish this Cantonese version from the original Cantonese MoCA version, we refer to it as the Hong Kong version of the Montreal Cognitive Assessment for the Visually Impaired (HKMoCA-VI).

The initial study of HKMoCA-VI involved 34 visually impaired elderly with cognitive impairment and 17 neurologically healthy elderly with visual impairment [20]. Researchers identified a cut-off score of 7 to differentiate dementia from mild cognitive impairment (MCI)

and normal controls, with a sensitivity and specificity of 100% and 65%, respectively. This cut-off score was notably lower than that of the original MoCA-BLIND (18 points) and slightly lower than the Malay version for the visually impaired elderly individuals (MoCA-BM-blind [21]), which had a cut-off score of 9. However, it should be noted that the education level of the elderly recruited in Fung [20] was generally low, with 91% of the participants being illiterate or having received six years of education or less. While the education level of MoCA-BM-blind participants was not reported, the mean years of education for healthy controls and dementia patients using MoCA were 13.3 years and 10 years, respectively. A larger sample size with a more diverse educational background may be necessary to develop HKMoCA-VI.

As the population ages, there is an increasing demand for quality long-term residential services for the elderly. Previous studies have shown that a majority of residents in nursing homes experience cognitive decline [22, 23]. A similar finding has been reported in Hong Kong, where about 57% of residents without visual impairment residing in nursing homes were found to have very mild to moderate dementia [24], as measured with Clinical Dementia Rating (CDR) and Mini-Mental State Examination (MMSE). Given that visual impairment may exacerbate cognitive decline, further investigation into the cognitive profile of visually impaired elderly residing in nursing homes is warranted.

The objectives of the present study were 1) to determine the cut-off score and percentile score of HKMoCA-VI and 2) to explore the concurrent validity of HKMoCA-VI with the Cantonese version of MMSE for elderly with visual impairment living in residential homes.

## Method and materials

### Study design

This cross-sectional study between 1 November 2019 and 31 January 2020 assessed the cognitive function of all elderly individuals residing in residential homes for the visually impaired who fulfilled the selection criteria. We then classified them into a dementia group and a non-dementia group according to their medical diagnoses.

### Participants

The study employed convenience sampling and conducted data collection in residential homes operated by the Hong Kong Society for the Blind (HKSB). Elderly individuals who met the following selection criteria were invited to sign a written consent form and participate in the study. Inclusion criteria were: 1) Chinese individuals aged 60 years or above, 2) speaking Cantonese as their primary language, 3) medically diagnosed with visual impairment (low vision of 20/200 or worse), 4) residing in a residential home, and 5) capable of following instructions. Exclusion criteria were: 1) living outside of residential homes, 2) not speaking Cantonese as their primary language, 3) medically diagnosed with depression, delirium, anxiety, or other mental illnesses, 4) not capable of following instructions, and 5) individuals or their guardians who are unable to provide written consent for participation in the study.

### Data collection procedures

The case therapists at residential homes assisted in screening the elderly to identify those meeting the selection criteria and guided them to sign the consent form. For individuals unable to comprehend the content of the consent form and thus incapable of making decisions, we sought consent from their guardians following a detailed explanation provided by the residential home staff. Subjects underwent assessment by a trained research assistant and trained HKSB staff using HKMoCA-VI and the CMMSE [25]. Assessments were conducted on the

same day but during different sessions (morning and afternoon). Demographic data, such as education level and medical diagnosis of dementia, were also collected for each subject.

## Statistical analysis

Statistical analysis utilized Statistical Package for the Social Sciences-Version 22 (SPSS, Inc., Chicago, IL, USA). The significance level was set at $p < 0.05$ (two-tailed).

Descriptive statistics were employed to analyze demographic characteristics. Percentile scores of HKMoCA-VI were calculated. A regression model investigated the relationship between HKMoCA-VI scores, age, and education level. A receiver operating characteristic (ROC) curve determined the cut-off score of HKMoCA-VI. The concurrent validity of HKMoCA-VI was assessed through a correlation coefficient with CMMSE.

## Ethical approval

Ethical approval was obtained from the Research Ethics Committee of the School of Medical and Health Sciences, Tung Wah College (approval number: RSS400130092019).

## Results

A total of 123 subjects (54 males and 69 females) participated in the study and completed cognitive assessments using HKMoCA-VI and CMMSE. Their ages ranged from 62 to 102 years, with a mean (standard deviation; S.D.) of 82.8 (9.7). Of these participants, 29.3% had a confirmed diagnosis of dementia. Their demographic characteristics are shown in Table 1.

The elderly attained mean scores of 12.3 (S.D. = 5.7) and 16.8 (S.D. = 7.2) on HKMoCA-VI and CMMSE, respectively. The Pearson correlation coefficient between HKMoCA-VI and CMMSE was.829 ($p < .001$). A regression model adjusted for age and level of education revealed an R-squared value of 0.272, indicating that age and level of education explained 27.2% of the variance in HKMoCA-VI scores. The model HKMoCA-VI = 31.33 + 0.653 (education level)– 0.255 (age) was statistically significant with $p < 0.001$. The unstandardized coefficients (B) for education level and age were statistically significant at $p = 0.017$ and $p < 0.001$, respectively. Subsequently, we calculated the percentiles of HKMoCA-VI scores by age group, education level, and their combination, as shown in Tables 2–4, respectively.

To determine the HKMoCA-VI cut-off score, we categorized the subjects based on their medical diagnosis into those with and without dementia. Subjects diagnosed with dementia were significantly older than those without (see Table 5). Additionally, the two groups significantly differed in their CMMSE and HKMoCA-VI scores (see Table 5).

Referring to Table 2, adjusting to the age group, 75% of subjects without a diagnosis of dementia scored above the 25[th] percentile on HKMoCA-VI. Conversely, 60% of subjects diagnosed with dementia scored below the 25[th] percentile.

The HKMoCA-VI cut-off scores for dementia, based on the medical diagnosis and the CMMSE cut-off score, were determined using the ROC curves, with results presented in Table 6. A cut-off score of 12 showed the highest sensitivity, specificity, and Youden's Index when using the CMMSE criterion for dementia. A cut-off score of 9 showed the highest sensitivity, specificity, and Youden's Index when based on medical diagnosis. The areas under the ROC curves were 0.93 and 0.807 (p < 0.001), respectively (see Fig 1).

## Discussion

The present study aimed to establish the cut-off score of HKMoCA-VI and evaluate the cognitive functions of visually impaired elderly residing in nursing homes. Cognitive function

**Table 1. Demographic characteristics of subjects.**

| | | Count (n = 123) | Percentage |
|---|---|---|---|
| Gender | | | |
| | Male | 54 | 43.9% |
| | Female | 69 | 56.1% |
| Age group | | | |
| | 60 to 69 years | 21 | 17.1% |
| | 70 to 79 years | 20 | 16.3% |
| | 80 to 89 years | 55 | 44.7% |
| | 90 years or above | 27 | 22.0% |
| Education level | | | |
| | No formal education | 31 | 25.2% |
| | Primary school or below | 38 | 30.9% |
| | Secondary 3 | 26 | 21.1% |
| | Secondary 5 or above | 28 | 22.8% |
| Residential home | | | |
| | Kowloon Home for the Aged Blind | 24 | 19.5% |
| | Jockey Club Tuen Mun Home for the Aged Blind | 39 | 31.7% |
| | Yuen Long Home for the Aged Blind (Jockey Club Yan Hong Building) | 25 | 20.3% |
| | Bradbury Care and Attention Home for the Aged Blind | 35 | 28.5% |
| Medical diagnosis of dementia | | | |
| | No | 87 | 70.7% |
| | Yes | 36 | 29.3% |
| | | Mean (n = 123) | Standard Deviation (S.D.) |
| Age | | 82.7 | 9.7 |
| HKMoCA-VI | | 12.3 | 5.7 |
| CMMSE | | 16.8 | 7.2 |

HKMoCA-VI, the Hong Kong version of the Montreal Cognitive Assessment for the visually impaired; CMMSE, the Cantonese version of the Mini-Mental State Examination.

assessment was conducted on 123 elderly individuals aged 60 years or above using cognitive screening tasks, namely the CMMSE and HKMoCA-VI. The former was used to determine the cut-off score and develop the concurrent validity of HKMoCA-VI. It was found that HKMoCA-VI effectively distinguished cognitive decline among visually impaired elderly,

**Table 2. Percentile of HKMoCA-VI score by age group.**

| Age group | Percentile | | | | | | | | | | |
|---|---|---|---|---|---|---|---|---|---|---|---|
| | 2 | 7 | 12.5 | 16 | 25 | 37.5 | 50 | 52.5 | 75 | 87.5 | 100 |
| 60–69 (n = 21) | 4 | 7 | 11 | 12 | 13 | 17 | 19 | 19 | 20 | 21 | 22 |
| 70–79 (n = 20) | 5 | 6 | 7 | 8 | 10 | 13 | 16 | 16 | 19 | 20 | 22 |
| 80–89 (n = 55) | 1 | 4 | 5 | 6 | 7 | 9 | 11 | 12 | 15 | 18 | 22 |
| 90 or above (n = 27) | 1 | 2 | 3 | 4 | 5 | 7 | 9 | 9 | 12 | 17 | 20 |
| All participants (N = 123) | 2 | 4 | 5 | 6 | 7 | 10 | 12 | 13 | 17 | 20 | 22 |

**Table 3. Percentile of HKMoCA-VI score by education level.**

| Education level | Percentile | | | | | | | | | | |
|---|---|---|---|---|---|---|---|---|---|---|---|
| | 2 | 7 | 12.5 | 16 | 25 | 37.5 | 50 | 52.5 | 75 | 87.5 | 100 |
| No formal education (n = 31) | 1 | 3 | 4 | 4 | 6 | 7 | 9 | 9 | 15 | 19 | 21 |
| Primary school or below (n = 38) | 1 | 3 | 4 | 4 | 6 | 8 | 10 | 10 | 15 | 19 | 22 |
| Secondary 3 (n = 26) | 5 | 6 | 10 | 11 | 12 | 13 | 14 | 15 | 20 | 21 | 22 |
| Secondary 5 or above (n = 28) | 5 | 6 | 9 | 9 | 11 | 14 | 15 | 15 | 18 | 20 | 21 |

**Table 4. Percentile of HKMoCA-VI score by age group and education level.**

| Age Group | Education level | Percentile | | | | | | | | | | |
|---|---|---|---|---|---|---|---|---|---|---|---|---|
| | | 2 | 7 | 12.5 | 16 | 25 | 37.5 | 50 | 52.5 | 75 | 87.5 | 100 |
| 60–69 | No formal education | 4 | 4 | 4 | 4 | 6 | 12 | 16 | 17 | 19 | | 19 |
| | Primary school or below | | | | | | | | | | | |
| | Secondary 3 | 13 | 13 | 13 | 14 | 18 | 20 | 20 | 20 | 21 | | 22 |
| | Secondary 5 or above | 10 | 10 | 10 | 11 | 13 | 16 | 17 | 18 | 21 | 21 | 21 |
| 70–79 | No formal education | 7 | 7 | 7 | 7 | 9 | 15 | 18 | 18 | 20 | | 20 |
| | Primary school or below | 9 | 9 | 9 | 9 | 10 | 11 | 14 | 14 | 19 | | 20 |
| | Secondary 3 | 5 | 5 | 5 | 6 | 11 | 13 | 16 | 16 | 21 | | 22 |
| | Secondary 5 or above | 6 | 6 | 6 | 6 | 8 | 14 | 15 | 15 | 16 | | 16 |
| 80–89 | No formal education | 1 | 1 | 4 | 5 | 6 | 8 | 10 | 10 | 15 | 18 | 21 |
| | Primary school or below | 3 | 3 | 4 | 4 | 5 | 7 | 8 | 8 | 15 | 15 | 22 |
| | Secondary 3 | 6 | 6 | 7 | 8 | 11 | 12 | 14 | 14 | 20 | 21 | 21 |
| | Secondary 5 or above | 5 | 5 | 7 | 9 | 9 | 12 | 14 | 14 | 16 | 18 | 18 |
| 90 or above | No formal education | 2 | 2 | 3 | 3 | 5 | 6 | 6 | 6 | 8 | 11 | 11 |
| | Primary school or below | 1 | 1 | 2 | 3 | 4 | 6 | 8 | 8 | 10 | 18 | 20 |
| | Secondary 3 | 12 | 12 | 12 | 12 | 12 | 12 | 13 | 13 | 16 | | 17 |
| | Secondary 5 or above | 9 | 9 | 9 | 9 | 9 | 12 | 15 | 15 | | | 19 |

showing a strong correlation with the CMMSE (r = 0.829; $p < 0.001$), demonstrating good concurrent validity. Furthermore, comparing cut-off scores of HKMoCA-VI based on dementia diagnosis versus CMMSE cut-off scores revealed better sensitivity, specificity, and a balanced approach (as indicated by Youden's index) for the latter. We, therefore, suggest that HKMoCA-VI is a clinically effective tool for cognitive screening in elderly with visual impairment, with an optimal cut-off score of 12, akin to the MoCA-22 for mild to moderate

**Table 5. Comparison of age and cognitive performance between groups.**

| | Group with Dementia diagnosis (n = 36) | | Group without Dementia diagnosis (n = 87) | | |
|---|---|---|---|---|---|
| | Mean | (S.D.) | Mean | (S.D.) | |
| Age | 87.9 | (6.61) | 80.6 | (10.10) | $p < 0.001$ |
| CMMSE | 13.0 | (6.59) | 18.4 | (6.82) | $p < 0.001$ |
| HKMoCA-VI | 8.0 | (4.40) | 14.1 | (5.27) | $p < 0.001$ |

**Table 6. HKMoCA-VI cut-off score determined by dementia diagnosis or CMMSE cut-off score.**

| Cut-off score | By Medical Diagnosis | | | By CMMSE cut-off score | | |
|---|---|---|---|---|---|---|
| | Sensitivity | Specificity | Youden's Index | Sensitivity | Specificity | Youden's Index |
| 7 | 85.10 | 50.00 | 0.351 | 98.21 | 44.80 | 0.430 |
| 8 | 83.91 | 61.10 | 0.45 | 98.21 | 52.20 | 0.505 |
| 9 | 79.31 | 72.22 | 0.515* | 96.43 | 62.69 | 0.591 |
| 10 | 73.56 | 75.00 | 0.486 | 92.86 | 68.66 | 0.615 |
| 11 | 67.82 | 80.56 | 0.484 | 91.07 | 77.61 | 0.687 |
| 12 | 63.22 | 83.33 | 0.466 | 89.29 | 83.58 | 0.729* |
| 13 | 57.47 | 86.11 | 0.436 | 82.14 | 86.57 | 0.687 |
| 14 | 51.72 | 91.67 | 0.434 | 75.00 | 91.04 | 0.660 |

dementia [18]. In addition, considering the influence of age and education on cognitive performance, healthcare professionals can utilize percentile scores stratified by age/education to assess cognitive status in older adults with visual impairment, particularly those in residential homes, for management planning.

We aimed to compare the psychometric properties of HKMoCA-VI with other MoCA versions. Its sensitivity and specificity values are slightly lower than that of HKMoCA, which has a sensitivity and specificity of 92.3% and 91.8%, respectively. While MoCA-BLIND demonstrates high sensitivity (94%) and specificity (98%), MoCA-BM-blind shows comparable sensitivity (86.8%) and specificity (72.7%) to HKMoCA-VI. Regarding cut-off scores, HKMoCA-VI aligns closely with the Spanish version (cut-off score of 12) and is more akin to MoCA-BM-blind (cut-off of 9) than MoCA (cut-off of 18). The difference in the participants' education

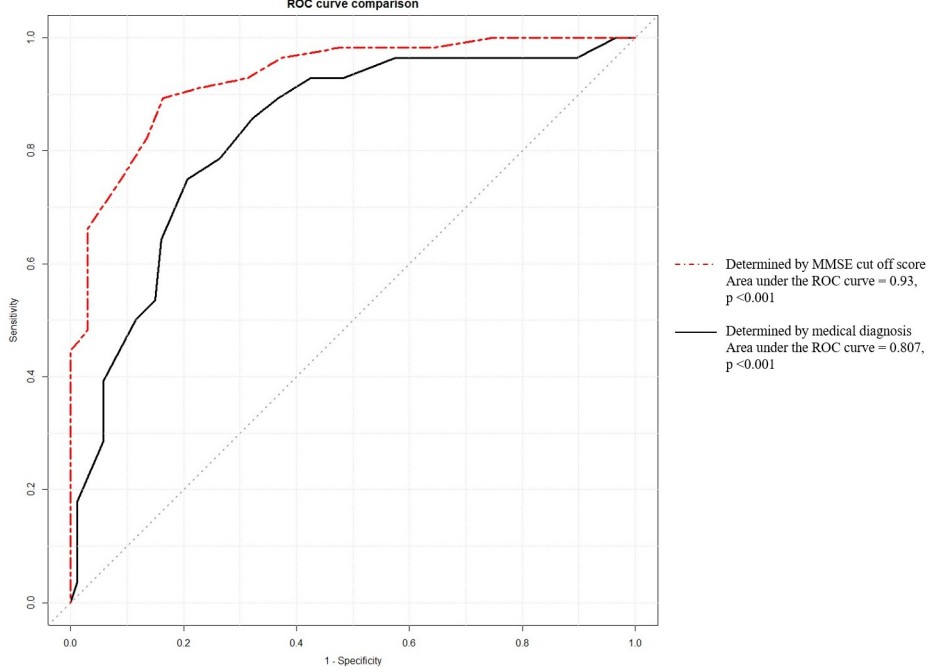

**Fig 1. ROC curves of HKMoCA-VI determined by dementia diagnosis and CMMSE cut-off score.**

levels in MoCA-BLIND (over 10 years) and HKMoCA-VI (8.4 years) may account for the difference in cut-off scores. Besides, MoCA-BLIND validation involved re-analyzing the existing data collected from the non-visually impaired population, while MoCA-BM-blind exclusively tested elderly with visual impairment for validation. Notably, in Fadzil et al. [21], participants without visual impairment attained a higher mean score than the visually impaired group (i.e., 18.76 versus 17.53), although the statistical significance of this difference was not reported. In sight of this, we propose that including visually impaired participants in test validation could better account for their performance and, hence, determine cut-off scores.

The availability of HKMoCA-VI is expected to enhance dementia identification awareness among nurses and other healthcare professionals, facilitating regular cognitive screening of elderly residents in residential homes.

Our study represents one of the pioneering investigations in Hong Kong exploring the general cognitive profile of visually impaired elderly residing in residential homes. Based on the percentile rank in Table 2, approximately half of the visually impaired elderly participants in the current study scored above the cut-off, offering insight into the overall cognitive functioning of elderly individuals living in residential homes for the visually impaired. The prevalence rate of dementia, as determined by HKMoCA-VI in the current study, appears to be slightly lower than that reported by Cheng et al. [24], although different cognitive measures were employed. For elderly individuals not exhibiting signs of cognitive decline, regular screening and interventions aimed at managing and preventing further deterioration of both visual impairment and cognitive functioning are essential. For visually impaired elderly experiencing cognitive decline, interventions such as cognitive stimulation and evidence-based therapy for dementia [26] should be adapted to accommodate their vision loss. Additionally, the use of technology, such as digitally enhanced visual and auditory stimuli delivered via virtual reality [27], could be considered.

### Limitations and suggestions for future research

The current study has several limitations. The first concerns the definition of the sample recruited for determining the cut-off scores of HKMoCA-VI. In some previous studies, the clinical and control groups were clearly defined (e.g., [19, 21]) through comprehensive neuropsychological assessments. However, due to the lack of cognitive assessment tools for older adults with visual impairment in Hong Kong, our study resorted to the CMMSE to determine the cut-off score. This yielded better sensitivity and specificity, especially given the under-recognition of dementia in long-term care homes in Hong Kong [24]. Another limitation pertains to the unavailability of data or test measures for a more comprehensive evaluation of the psychometric properties of HKMoCA-VI. Test-retest reliability and internal consistency were not assessed. Including these measures would undoubtedly strengthen the test. Additionally, unlike the MoCA-BLIND and HKMoCA, our study did not consider evaluating cut-off scores for individuals with mild cognitive impairment (MCI). Future research on identifying cut-off scores for visually impaired elderly with MCI is strongly recommended. Moreover, approximately 25% of participants received no formal education, and the impact of education on their cognitive performance remained inconclusive. Similarly, age and education level, known to be associated with cognitive functions in the elderly [28, 29], were not thoroughly considered in determining the cut-off score. Employing age and education-corrected cut-off scores, as adopted in the 5-minute protocol of HKMoCA [17], may provide a more accurate screening tool for the population.

Fung [20] raised a significant concern regarding the recruitment of visually impaired participants from nursing homes. Previous studies [30, 31] highlighted the association between

place of residence and cognitive decline. Moreover, the elderly in our study had severe visual impairment. These sample characteristics limit the generalizability of our findings to community-dwelling elderly with varying degrees of visual impairment in Hong Kong. In summary, we recommend validating HKMoCA-VI with a larger sample, encompassing community-dwelling and nursing home-residing elderly with different ages and education levels.

## Conclusion

HKMoCA-VI proves to be a clinically viable tool for cognitive screening among Cantonese-speaking elderly individuals. The performance of visually impaired elderly residing in residential homes on HKMoCA-VI may offer insights into the elderly's overall cognitive functioning, providing valuable information for the management and service planning for this population.

## Acknowledgments

We extend our gratitude to the elderly residents and the staff members of the elderly homes at the Hong Kong Society for the Blind, and the research assistants for their participation and assistance in this study.

## Author Contributions

**Conceptualization:** Calvin Chi Kong Yip, Winsy Wing Sze Wong, Calvin Pak Wing Cheng, Armstrong Tat San Chiu.

**Data curation:** Calvin Chi Kong Yip, Winsy Wing Sze Wong.

**Formal analysis:** Calvin Chi Kong Yip, Armstrong Tat San Chiu.

**Investigation:** Calvin Chi Kong Yip, Winsy Wing Sze Wong, Calvin Pak Wing Cheng, Armstrong Tat San Chiu.

**Methodology:** Calvin Chi Kong Yip, Armstrong Tat San Chiu.

**Project administration:** Calvin Chi Kong Yip, Armstrong Tat San Chiu.

**Resources:** Calvin Chi Kong Yip, Armstrong Tat San Chiu.

**Software:** Calvin Chi Kong Yip, Armstrong Tat San Chiu.

**Supervision:** Calvin Chi Kong Yip, Armstrong Tat San Chiu.

**Validation:** Calvin Chi Kong Yip, Winsy Wing Sze Wong, Armstrong Tat San Chiu.

**Visualization:** Calvin Chi Kong Yip, Armstrong Tat San Chiu.

**Writing – original draft:** Calvin Chi Kong Yip, Winsy Wing Sze Wong.

**Writing – review & editing:** Calvin Chi Kong Yip, Winsy Wing Sze Wong, Armstrong Tat San Chiu.

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
