## [Decision Letter · Decision Letter 0]

9 Feb 2024

PONE-D-23-42809The Hong Kong version of Montreal Cognitive Assessment for the visually impaired (HKMoCA-VI): Its cutoff and a survey on the cognitive functioning of visually impaired elderly in residential homesPLOS ONE

Dear Dr. Wong,

Thank you for submitting your manuscript to PLOS ONE. After careful consideration, we feel that it has merit but does not fully meet PLOS ONE’s publication criteria as it currently stands. Therefore, we invite you to submit a revised version of the manuscript that addresses the points raised during the review process. Two reviewers made significant comments, and they are reasonable. Please improve the manuscript following the comments.

We look forward to receiving your revised manuscript.

Kind regards,

Ryota Sakurai, Ph.D.

Academic Editor

PLOS ONE

2. Please describe in your methods section how capacity to provide consent was determined for the participants in this study. Please also state whether your ethics committee or IRB approved this consent procedure. If you did not assess capacity to consent please briefly outline why this was not necessary in this case.

4. We notice that your supplementary figures are uploaded with the file type 'Figure'. Please amend the file type to 'Supporting Information'. Please ensure that each Supporting Information file has a legend listed in the manuscript after the references list.

Reviewers' comments:

Reviewer's Responses to Questions

**Comments to the Author**

1. Is the manuscript technically sound, and do the data support the conclusions?

Reviewer #1: Yes

Reviewer #2: Partly

2. Has the statistical analysis been performed appropriately and rigorously? 

Reviewer #1: No

Reviewer #2: No

3. Have the authors made all data underlying the findings in their manuscript fully available?

Reviewer #1: Yes

Reviewer #2: Yes

4. Is the manuscript presented in an intelligible fashion and written in standard English?

Reviewer #1: No

Reviewer #2: Yes

5. Review Comments to the Author

Reviewer #1: This paper represents a major effort to establish the cutoff score for the Hong Kong Version of the Montreal Cognitive Assessment for the visually impaired and to assess the cognitive performance of nursing home elderly residents with visual impairment in Hong Kong. I believe that the goal of the study is worthwhile and appropriate for the readership of the PLOS ONE. The study is providing important data that can be applied in clinical practice. However, major concerns should be addressed before resubmission.

1. The manuscript should be checked by professional editing service before resubmission.

2. The introduction section is long. While comprehensive, the introduction could be more impactful by focusing on the most essential background information.

3. The quality of the image in figure 1 is poor and should be improved.

4. The Cantonese version of the Mini Mental State Examination was described in the manuscript in different ways as CMMSE-VI, CMMSE, The modified CMMSE. The authors need to unify the term used to describe the tool throughout the manuscript.

5. Regarding the methodology section: there are many major concerns as

o While restricting the study to individuals with severe visual impairment simplifies the analysis by minimizing confounding factors, it also limits the generalizability of the findings. This point should be acknowledged in the limitations section.

o Another limitation of the study was including those with no information regarding their education level. 31 cases that comprises 25.2% of the sample recruited.

o Restricting the study to cases in nursing homes will prevent results generalizability to elderly within community, however; this limitation was highlighted in the limitations.

o The absence of data on depression and delirium, known to affect memory and attention, makes it difficult to draw definitive conclusions about the cognitive abilities of the cases. Were the cases with depression and /or delirium excluded from the study?

o The Cantonese version of the Mini Mental State Examination (as cited in reference 27) recommended a cut-off score of 19/20 that yielded a high sensitivity rate of 97.5 and an equally good specificity of 97.3. This appeared to be the best cut-off score for their sample of subjects. Moreover, those cut off scores were obtained without adjusting for visual impairment thus can’t be applied to your sample.

o Linear regression would adjust better for age and educations than using the percentiles, then it would be better to add tables 2,3,4 as supplementary data.

Reviewer #2: Dear Authors

I read your work entitled “The Hong Kong version of Montreal Cognitive Assessment for the visually impaired (HKMoCA-VI): Its cutoff and a survey on the cognitive functioning of visually impaired elderly in residential homes” and here I enclose my recommendations to you:

1. Please have the text checked for language issues, since several mistakes have been tracked. A native speaker of English or a professional Editing company would help towards this.

2. Abstract General comments: The Abstract need a reconstruction and Methods section and the results section.

3. The “Introduction” section: It would be beneficial for the authors to include more precise information in the introduction about the gaps of the existing knowledge and how this study attempts to fill them. An augmentation of the existing literature in the text would benefit more for the Introduction and it will strengthen the rational of this study. Additionally, I suggest the Authors to focus more on the HKMoCA-VI and why it is so important.

4. The “Methods” section: The Authors report criteria that are not based on the existing literature. I suggest them to elaborate this.

5. The “Methods” section: It is strange that the Author do not report which groups severed as clinical and the controls in this study.

6. The “Discussion” section: The Authors are developing this section according to their results and connect them in an adequate manner to the existing knowledge but as we can see most of the studies used a control group that in this study is not clearly reported. I suggest the Authors to address that as well. This will help the Authors to strengthen more their work.

Thank you.

6. PLOS authors have the option to publish the peer review history of their article (what does this mean?). If published, this will include your full peer review and any attached files.

Reviewer #1: **Yes: **Doha Rasheedy

Reviewer #2: No

---

## [Author Response · Author response to Decision Letter 0]

8 Apr 2024

Response: The format and file naming of the manuscript has been checked and revised. 

2. Please describe in your methods section how capacity to provide consent was determined for the participants in this study. Please also state whether your ethics committee or IRB approved this consent procedure. If you did not assess capacity to consent please briefly outline why this was not necessary in this case.

Response: Capacity to provide consent was determined by their comprehension ability of the information provided when written consent was sought. For those who were not fit to provide consent themselves, their guardians were invited to give consent upon explanation given by the staff of the residential homes. The information was included on p. 9 (lines 146-150). Information on ethical approval is included (p. 10, lines 167-169).

Response: The data availability statement has been added (p. 20, lines 320-321). 

4. We notice that your supplementary figures are uploaded with the file type 'Figure'. Please amend the file type to 'Supporting Information'. Please ensure that each Supporting Information file has a legend listed in the manuscript after the references list.

Response: The file type of the ‘Figure’ has been amended with the legend listed in the manuscript. 

Reviewers' comments:

Reviewer's Responses to Questions

Comments to the Author

1. Is the manuscript technically sound, and do the data support the conclusions?

Reviewer #1: Yes

Reviewer #2: Partly

2. Has the statistical analysis been performed appropriately and rigorously?

Reviewer #1: No

Reviewer #2: No

3. Have the authors made all data underlying the findings in their manuscript fully available?

Reviewer #1: Yes

Reviewer #2: Yes

4. Is the manuscript presented in an intelligible fashion and written in standard English?

Reviewer #1: No

Reviewer #2: Yes

5. Review Comments to the Author

Reviewer #1: This paper represents a major effort to establish the cutoff score for the Hong Kong Version of the Montreal Cognitive Assessment for the visually impaired and to assess the cognitive performance of nursing home elderly residents with visual impairment in Hong Kong. I believe that the goal of the study is worthwhile and appropriate for the readership of the PLOS ONE. The study is providing important data that can be applied in clinical practice. However, major concerns should be addressed before resubmission.

1. The manuscript should be checked by professional editing service before resubmission.

Response: The manuscript has been checked by professional editing service before resubmission. 

2. The introduction section is long. While comprehensive, the introduction could be more impactful by focusing on the most essential background information.

Response: The introduction section has been revised to focus on the development of HKMoCA-VI and the rationales (p. 5, lines 79-93), with the implications of findings stressed in Discussion (p. 15, lines 225-226, 231-234). 

3. The quality of the image in figure 1 is poor and should be improved.

Response: The quality of the image of Figure 1 has been improved and re-submitted. 

4. The Cantonese version of the Mini Mental State Examination was described in the manuscript in different ways as CMMSE-VI, CMMSE, The modified CMMSE. The authors need to unify the term used to describe the tool throughout the manuscript.

Response: The term ‘CMMSE’ has been used consistently throughout the manuscript. 

5. Regarding the methodology section: there are many major concerns as

o While restricting the study to individuals with severe visual impairment simplifies the analysis by minimizing confounding factors, it also limits the generalizability of the findings. This point should be acknowledged in the limitations section.

Response: This point has been acknowledged in the limitations section (p. 19, lines 304-306). 

o Another limitation of the study was including those with no information regarding their education level. 31 cases that comprises 25.2% of the sample recruited.

Response: This point has been addressed in the limitation (p. 18, lines 293-295). 

o Restricting the study to cases in nursing homes will prevent results generalizability to elderly within community, however; this limitation was highlighted in the limitations.

Response: Thanks for raising this point out, which has been included as one of the limitations. 

o The absence of data on depression and delirium, known to affect memory and attention, makes it difficult to draw definitive conclusions about the cognitive abilities of the cases. Were the cases with depression and /or delirium excluded from the study?

Response: We excluded elderly with a diagnosis of depression, delirium or other mental illness (p. 9, lines 141-142). 

o The Cantonese version of the Mini Mental State Examination (as cited in reference 27) recommended a cut-off score of 19/20 that yielded a high sensitivity rate of 97.5 and an equally good specificity of 97.3. This appeared to be the best cut-off score for their sample of subjects. Moreover, those cut off scores were obtained without adjusting for visual impairment thus can’t be applied to your sample.

Response: The method for determining the HKMoCA-VI has been explained (p. 10, lines 157-165, p. 11, lines 182-190, 196-213). 

o Linear regression would adjust better for age and educations than using the percentiles, then it would be better to add tables 2,3,4 as supplementary data.

Response: Linear regression has been conducted to account for the effects of age and education on the scores (p. 11, lines 182-190). We would still keep tables 2-4 in the main text for the clinicians’ easier reference. 

Reviewer #2: Dear Authors

I read your work entitled “The Hong Kong version of Montreal Cognitive Assessment for the visually impaired (HKMoCA-VI): Its cutoff and a survey on the cognitive functioning of visually impaired elderly in residential homes” and here I enclose my recommendations to you:

1. Please have the text checked for language issues, since several mistakes have been tracked. A native speaker of English or a professional Editing company would help towards this.

Response: The manuscript has been checked for language accuracy. 

2. Abstract General comments: The Abstract need a reconstruction and Methods section and the results section.

Response: The abstract has been reconstructed (p. 2, lines 25-31, 37-39). 

3. The “Introduction” section: It would be beneficial for the authors to include more precise information in the introduction about the gaps of the existing knowledge and how this study attempts to fill them. An augmentation of the existing literature in the text would benefit more for the Introduction and it will strengthen the rational of this study. Additionally, I suggest the Authors to focus more on the HKMoCA-VI and why it is so important.

Response: The Introduction section has been revised to focus on HKMoCA-VI (p. 5, lines 79-93; p. 7, lines 115-118). 

4. The “Methods” section: The Authors report criteria that are not based on the existing literature. I suggest them to elaborate this.

Response: Inclusion and exclusion criteria have been elaborated (p. 8, lines 127-144). 

5. The “Methods” section: It is strange that the Author do not report which groups severed as clinical and the controls in this study.

Response: We attempted to classify the participants into dementia group and non-dementia group according to their medical diagnosis (p. 8, lines120-125). Nevertheless, due to the under-recognition of dementia in Hong Kong while CMMSE has been used to determine cut-off of HKMoCA-VI. This point has been discussed in the Discussion section (p. 17-18, lines 278-285). 

6. The “Discussion” section: The Authors are developing this section according to their results and connect them in an adequate manner to the existing knowledge but as we can see most of the studies used a control group that in this study is not clearly reported. I suggest the Authors to address that as well. This will help the Authors to strengthen more their work.

Response: This point has been addressed in the Discussion section (p. 17-18, lines 278-285).

---

## [Decision Letter · Decision Letter 1]

7 May 2024

PONE-D-23-42809R1The Hong Kong version of Montreal Cognitive Assessment for the visually impaired (HKMoCA-VI): Its cut-off and a survey on the cognitive functioning of visually impaired elderly in residential homesPLOS ONE

Dear Dr. Wong,

Thank you for submitting your manuscript to PLOS ONE. After careful consideration, we feel that it has merit but does not fully meet PLOS ONE’s publication criteria as it currently stands. Therefore, we invite you to submit a revised version of the manuscript that addresses the points raised during the review process.

We look forward to receiving your revised manuscript.

Kind regards,

Ryota Sakurai, Ph.D.

Academic Editor

PLOS ONE

Journal Requirements:

**Additional Editor Comments:**

Thank you for your effort in improving your manuscript. Reviewer #1 made a minor comment. Since they are reasonable comments, please respond to them.

Reviewers' comments:

Reviewer's Responses to Questions

**Comments to the Author**

1. If the authors have adequately addressed your comments raised in a previous round of review and you feel that this manuscript is now acceptable for publication, you may indicate that here to bypass the “Comments to the Author” section, enter your conflict of interest statement in the “Confidential to Editor” section, and submit your "Accept" recommendation.

Reviewer #1: (No Response)

Reviewer #2: All comments have been addressed

2. Is the manuscript technically sound, and do the data support the conclusions?

Reviewer #1: Yes

Reviewer #2: Yes

3. Has the statistical analysis been performed appropriately and rigorously? 

Reviewer #1: Yes

Reviewer #2: Yes

4. Have the authors made all data underlying the findings in their manuscript fully available?

Reviewer #1: Yes

Reviewer #2: Yes

5. Is the manuscript presented in an intelligible fashion and written in standard English?

Reviewer #1: No

Reviewer #2: Yes

6. Review Comments to the Author

Reviewer #1: This paper represents a major effort to establish the cutoff score for the Hong Kong Version of the Montreal Cognitive Assessment for the visually impaired and to assess the cognitive performance of nursing home elderly residents with visual impairment in Hong Kong. I believe that the goal of the study is worthwhile and appropriate for the readership of the PLOS ONE. However, minor changes are required.

• The introduction includes some redundant information. Consider streamlining it to focus on the core research question and the rationale for the study.

• The manuscript would benefit from some English editing to ensure clarity and flow. Consider using a professional editing service or collaborating with a colleague with strong English language skills.

• Currently, the inclusion and exclusion criteria are presented in separate bullet points. For improved readability, consider combining them into a single, well-structured paragraph. This will allow you to explain the rationale behind each criterion more effectively.

Reviewer #2: Dear authors,

Thank you for addressing all my comments. Please have a final check again for English.

Thank you.

7. PLOS authors have the option to publish the peer review history of their article (what does this mean?). If published, this will include your full peer review and any attached files.

Reviewer #1: **Yes: **Doha Rasheedy

Reviewer #2: No

---

## [Author Response · Author response to Decision Letter 1]

22 May 2024

1. We note your current Data Availability statement is: All datasets will be available from open-access repository (e.g., OSF, Harvard Dataverse) upon manuscript acceptance.;

Tick here if the URLs/accession numbers/DOIs will be available only after acceptance of the manuscript for publication so that we can ensure their inclusion before publication."

In the online submission form, you indicated that your data will be submitted to a repository upon acceptance. We strongly recommend all authors deposit their data before acceptance, as the process can be lengthy and hold up publication timelines. Please note that, though access restrictions are acceptable now, your entire minimal dataset will need to be made freely accessible if your manuscript is accepted for publication. This policy applies to all data except where public deposition would breach compliance with the protocol approved by your research ethics board. If you are unable to adhere to our open data policy, please kindly revise your statement to explain your reasoning and we will seek the editor's input on an exemption.

Response: The data availability statement 'The raw data in the validation process is available at https://osf.io/62gwf/files/osfstorage/6614a02ac053943299b4dd01' , which is included in the mauscript, has been updated in the online submission form. The corresponding box 'Tick here if the URLs/accession numbers/DOIs will be available only after acceptance of the manuscript for publication so that we can ensure their inclusion before publication." has been checked. The dataset has been uploaded and will be made public after manuscript acceptance. 

Thanks you very much. 

Regards,

Winsy Wong

---

## [Decision Letter · Decision Letter 2]

30 May 2024

The Hong Kong version of Montreal Cognitive Assessment for the Visually Impaired (HKMoCA-VI): Proposed cut-off and cognitive functioning survey of visually impaired elderly in residential homes

PONE-D-23-42809R2

Dear Dr. Wong,

We’re pleased to inform you that your manuscript has been judged scientifically suitable for publication and will be formally accepted for publication once it meets all outstanding technical requirements.

Kind regards,

Ryota Sakurai, Ph.D.

Academic Editor

PLOS ONE

Additional Editor Comments (optional):

The manuscript improved with comments from the reviewers. Thanks for providing important information to PLOS ONE.

Reviewers' comments:

Reviewer's Responses to Questions

**Comments to the Author**

1. If the authors have adequately addressed your comments raised in a previous round of review and you feel that this manuscript is now acceptable for publication, you may indicate that here to bypass the “Comments to the Author” section, enter your conflict of interest statement in the “Confidential to Editor” section, and submit your "Accept" recommendation.

Reviewer #1: All comments have been addressed

2. Is the manuscript technically sound, and do the data support the conclusions?

Reviewer #1: Yes

3. Has the statistical analysis been performed appropriately and rigorously? 

Reviewer #1: Yes

4. Have the authors made all data underlying the findings in their manuscript fully available?

Reviewer #1: Yes

5. Is the manuscript presented in an intelligible fashion and written in standard English?

Reviewer #1: Yes

6. Review Comments to the Author

Reviewer #1: (No Response)

7. PLOS authors have the option to publish the peer review history of their article (what does this mean?). If published, this will include your full peer review and any attached files.

Reviewer #1: No

---

## [Editor Report · Acceptance letter]

18 Jun 2024

PONE-D-23-42809R2 

PLOS ONE

Dear Dr. Wong, 

I'm pleased to inform you that your manuscript has been deemed suitable for publication in PLOS ONE. Congratulations! Your manuscript is now being handed over to our production team.

Kind regards, 

on behalf of

Dr. Ryota Sakurai 

Academic Editor

PLOS ONE